# Antimicrobial Effect of the Amniotic Membrane Isolated and Associated with Photodynamic Therapy

**DOI:** 10.3390/jfb14030151

**Published:** 2023-03-08

**Authors:** Amanda Cerquearo Rodrigues dos Santos, Guilherme Rodrigues Teodoro, Juliana Ferreira-Strixino, Luciana Barros Sant’Anna

**Affiliations:** 1Center for Studies and Microbiological Analysis (CEAM)–Golden Technology, Av. Shishima Hifumi, São José dos Campos 2911, SP, Brazil; 2Photobiology Applied to Health (PhotoBioS), Research and Development Institute, University of Vale do Paraiba, Av. Shishima Hifumi, São José dos Campos 2911, SP, Brazil; 3Histology and Regenerative Therapy, Research and Development Institute, University of Vale do Paraiba, Av. Shishima Hifumi, São José dos Campos 2911, SP, Brazil

**Keywords:** amniotic membrane, phthalocyanine, photosensitizer, antimicrobial effect, photodynamic therapy

## Abstract

Microbial control through alternative therapies, such as the amniotic membrane (AM) and antimicrobial photodynamic therapy (aPDT), has been gaining prominence with the advancement of bacterial resistance to conventional treatments. This study aimed to evaluate the antimicrobial effect of AM isolated and associated with aPDT using the PHTALOX^®^ as a photosensitizer (PS) against *Staphylococcus aureus* and *Pseudomonas aeruginosa* biofilms. The groups studied were: C+; L; AM; AM+L; AM+PHTX; and AM+aPDT. The irradiation parameters were 660 nm, 50 J.cm^−2^, and 30 mW.cm^−2^. Two independent microbiological experiments were carried out in triplicate, and the results were analyzed by CFU/mL counting and a metabolic activity test, both statistically analyzed (*p* < 0.05). The integrity of the AM was verified after the treatments by a scanning electron microscope (SEM). The groups AM, AM+PHTX, and, mainly, AM+aPDT showed a statistical difference when compared to C+ regarding the decrease in CFU/mL and metabolic activity. SEM analysis showed significant morphological alterations in the AM+PHTX and AM+aPDT groups. The treatments with AM isolated or associated with PHTALOX^®^ were adequate. The association had potentiated the biofilm effect, and the morphological differences presented by AM after treatment did not hinder its antimicrobial effect, encouraging its use in biofilm formation locals.

## 1. Introduction

The human amniotic membrane (AM), the inner layer of the fetal membranes, discarded after delivery along with the placenta, is a tissue formed by cells with the potential to be stem cells and has great importance and a wide area of application due to its characteristics that include beneficial properties such as: low immunogenicity, anti-inflammatory properties, antiangiogenic properties, anti-fibrotic properties, pro-regenerative properties, and even antimicrobial potential [1,2,3,4]. The AM has been studied and used in various conditions, with its main applications being in the repair of the mucosa of the oral cavity [5], the cornea [6], and areas of the skin [7], as a scaffold or scaffold for tissue engineering [8], and antimicrobial purposes [9].

Schuerch et al. (2020) evaluated MA transplantation for the treatment of corneal ulcers, obtaining significantly positive results in 70% of patients, who achieved epithelial closure of the corneal ulcer with a single transplant procedure (21% in less than 1 month, 40% within 1–3 months, and 9% within 3–6 months). Another important finding of this study revealed that the highest closure rates are due to bacterial ulcers, herpetic ulcers, and neurotrophic ulcers (80%, 85%, and 93%, respectively). These results demonstrated the regenerative property of AM, as well as its antimicrobial capacity, presenting itself as a valuable non-conventional treatment option for achieving the healing of corneal epithelial wounds [10]. The study by Ferng et al. (2016) treated AM with a postoperative hospital infection, followed by the implantation of a left ventricular assist device, in a 66-year-old patient with a history of ischemic cardiomyopathy, chronic systolic heart failure, chronic coronary disease, arterial and renal arteries, and type 2 diabetes mellitus. The patient acquired coagulase-negative *Sthapylococcus* and *Candida albicans* after implantation, and the use of broad-spectrum antibiotics resulted in worsening of the infection. Thus, AM was used as a last resort treatment, and within 6 weeks, the wound was clear and the infection appeared to be resolved. After the resolution of the infection, a wound site grafting procedure with latissimus dorsi muscle tissue was performed, and the patient made a full recovery [11].

Currently, the class of drugs that is at the top of the list of the most prescribed medications in the world is antibiotics, a circumstance that may be responsible for the development of bacterial resistance [12]. Antibiotics are drugs, natural or synthetic, that have the ability to prevent the multiplication or cause the death of fungi and/or bacteria [13,14]. In recent decades, the discovery and development of antibiotics and antimicrobial products has revolutionized the treatment of bacterial infections, providing a sharp drop in mortality and faster and more effective treatment for the patient [14,15]. Bacterial resistance is a natural consequence of bacterial cell adaptation to exposure to antibiotics. However, the indiscriminate and improper use of this drug facilitates the acquisition of resistance mechanisms due to the increased exposure of the microorganism to these drugs [16]. According to the WHO, about 50% of antibiotic prescriptions are considered inadequate for the necessary treatment [17], and the resistance acquired by these microorganisms to antibiotics is irreversible. This condition is aggravated by selective pressure, considered one of the main contributing factors to microbial resistance [18].

With the increase in bacterial resistance due to the indiscriminate and inappropriate use of antibiotics, the interest in carrying out alternative therapies, such as the use of the amniotic membrane and the study of products with an antimicrobial effect, has been gaining prominence in the scientific literature, so new molecules and potential techniques have been studied. Phthalocyanines, traditionally known for being synthetic dyes, stand out as an example of these molecules, since they have achieved great importance in the microbiological area due to their promising antimicrobial properties [19,20]. Thus, the antimicrobial verification of the joint action of AM with antimicrobial products becomes a promising alternative for controlling microorganisms that cause infections [21].

Among the possibilities for applying such alternatives, the verification of their effect against bacteria often associated with wound infections stands out, since studies have shown that one of the main reasons for failure in the treatment of wounds and the evolution to a chronic wound is bacterial infection [22,23,24,25,26]. Among the most common bacteria isolated from chronic ulcers, *Staphylococcus aureus* stands out, usually detected in the upper portion of the wound, along with *Pseudomonas aeruginosa*, located in deeper regions, often in its biofilm state and resistant to conventional antimicrobial therapies [27,28,29,30,31]. Thus, it is possible to apply these alternative therapies against these bacteria in their biofilm state since this aggregation directly impacts their virulence due to the limitation of the penetration of antibiotics into the matrix, changes in the growth rate of micro-organisms that make up the biofilm, and physiological changes, which include the expression of possible resistance genes and resistance to the host’s immune response [32].

Another well-known alternative in the control of microorganisms is Antimicrobial Photodynamic Therapy (aPDT), a treatment methodology based on the use of photosensitizers (PS), where the PS is excited by a light source at an appropriate wavelength, transferring energy or electrons to molecular oxygen-forming Reactive Oxygen Species (ROS), responsible for causing irreversible structural damage to bacterial cells, such as the inactivation and death of these microorganisms [33]. Since then, this model of therapy has been widely studied and used as a form of treatment, with promising results. In 2020, the study by Yang et al., induced wounds infected by *P. aeruginosa* in sixty mice, performing treatment with PDT using 5-aminolevulinic acid (ALA) as a photosensitizer. The group treated with PDT-ALA was significantly lower in relation to the CFU/g count when compared to the control group (*p* < 0.05), demonstrating its antibacterial effect. In addition, a histological analysis was performed, where it was possible to observe the promotion of wound healing in the skin, with the formation of granulation tissue, angiogenesis, regeneration, and collagen remodeling [34].

In this context, the association of these alternative antimicrobial therapies becomes important in enhancing the results, creating a synergistic antimicrobial effect, taking into account that many of these microorganisms are becoming resistant to conventional treatments [35,36]. Therefore, the objective of the present study was to evaluate the antimicrobial effect of the amniotic membrane isolated and associated with photodynamic therapy using the PHTALOX^®^ solution as a photosensitizer against bacterial biofilms.

## 2. Materials and Methods

### 2.1. Placenta Collection

This study was submitted and approved by the research ethics committee of the University of Vale do Paraíba (UNIVAP) under the number 5.200.827/CEP/2021.

To verify the antimicrobial effect of the AM, five human placentas were obtained from elective cesarean sections of women with normal pregnancies after prior informed consent of the parturient and after checking the inclusion criteria (gestational age equal to or greater than 37 weeks; negative serology for Syphilis, Human Immunodeficiency Virus and Hepatitis B and C; pregnant women without flu symptoms in the last 14 days prior to placental collection; pregnant women with no history of COVID-19 infection). The placentas were placed in a sterile vat and then in a sterile plastic bag, under refrigeration at around 8 °C, with the coded identification of the donor. The placentas containing the fetal membranes were transported to the “Histology and Regenerative Therapy” laboratory of the Research and Development Institute (IP&D) of UNIVAP inside a thermal box under refrigeration at around 10 °C and isolated from the external environment to avoid possible damage.

### 2.2. Amniotic Membrane Processing

The procedures for AM processing were performed under sterile conditions inside a laminar flow hood. The umbilical cord was cut, the AM was manually separated from the chorion and completely washed with saline solution containing 100 U.mL^−1^ of penicillin (Gibco^tm^, Grand Island, NY, USA) and 100 mg.mL^−1^ of streptomycin and amphotericin B (Gibco^tm^, Grand Island, NY, USA) until it acquired a transparent aspect. After that, the AM was extensively washed with only a physiological solution to ensure that no antibiotic residues interfered with the tests. Regions with ruptures or with a gelatinous consistency were discarded, as well as areas with strongly adhered blood clots. Subsequently, the reflected amnion (RA) part of the AM was cut into rectangular fragments ± 1.5 × 2.0 cm and marked with an “L”-shaped cut on the right and upper part, with the mesenchymal side (matte side) facing down. The fragments were stored using the cryopreservation method in 50 mL tubes containing Dulbecco’s Modified Eagle Medium (DMEM, Sigma-Aldrich, St. Louis, MO, USA)/glycerol 1:1 at –80 °C for 10 days.

### 2.3. Phtalox^®^

A commercial phthalocyanine (PHTALOX^®^) was used as the PS, and an octavalent molecule was derived from phthalocyanine with a metallic iron core, batch 048/21. About 0.01 g of the powdered substance was dissolved in 1 mL of a 0.3% NaOH solution, resulting in a final solution with 1% PHTALOX^®^.

### 2.4. Microorganisms

For the study, standard strains of *S. aureus* American Type Culture Collection (ATCC) 6538 and *P. aeruginosa* ATCC 15442 were used, acquired from Microbiologics^®^ and provided by the company Golden Technology^®^. These were maintained in Trypticase Soy Broth (TSB, Merck, Darmstadt, Germany) with 20% glycerol between −18 and −20 °C, subcultured in Trypticase Soy Agar (TSA, Merck, Darmstadt, Germany), using up to the fifth passage at most. Bacteria were cultured 24 h before the experiment was carried out. Sequentially, an isolated colony of the microorganism was inoculated into 10 mL of Brain Heart Infusion Broth (BHI, Merck, Darmstadt, Germany) and incubated for 24 h at 37 °C under agitation in a shaker (Marconi, MA420) at 200 rotations per minute (rpm). The purity and viability of the cultures were also tested, according to strict laboratory biosafety standards.

#### 2.4.1. Inoculum Preparation

Inoculums containing about 10^8^ cells/mL were prepared from bacterial suspensions, grown in TSB. For this purpose, optical density (OD) values of 0.230 for *S. aureus* and 0.365 for *P. aeruginosa* were obtained in a spectrophotometer (Micronal, B582) with a wavelength of 475 nm.

#### 2.4.2. Biofilm Assembly

The biofilm was formed in 24-well plates (TPP Techno Plastic Products AG, Trasadingen, Switzerland) by pipetting 200 µL of the prepared inoculum into 1.8 mL of BHI broth, as schematized in Figure 1. These plates were incubated at 37 °C for 48 h under agitation in a shaker at 200 rpm, and the BHI culture medium was changed after 24 h.

### 2.5. Experimental Groups

Two independent experiments were carried out in triplicate, with the experimental groups divided into: Positive Control Group (C+), Light Group (L), AM Group (AM), AM and Light Group (AM+L), AM and PHTALOX^®^ solution Group (AM+PHTX), and AM and aPDT Group (AM+aPDT), as shown in Table 1.

### 2.6. Amniotic Membrane Analysis

After 10 days of cryopreservation, the 50 mL tubes containing the AM fragments in the DMEM/glycerol medium were removed from the −80 °C freezer and left at room temperature until complete defrosting. Three washes with a three-minute duration each in phosphate-buffered saline (PBS, LGC Biotecnologia, SP, Brazil) were performed sequentially, and the fragments were cut in a circular shape that was ± 1.5 cm in diameter. After the growth of the biofilm, the culture medium was removed from the wells and washed with 2 mL of PBS. Soon after, 200 µL of PBS was added, and the AM fragment was processed with the mesenchymal side facing the bottom of the plate containing the biofilm. The AM stayed in contact with the biofilm for 24 h at 37 °C in a humid chamber (AM Group). For the C+ group, the same process was performed, but with the absence of the AM fragment.

### 2.7. Analysis of the Amniotic Membrane Associated with Phtalox^®^ Solution

Initially, the same procedure described in item 2.6 was performed. However, after 24 h of contact with the AM, 1 mL of the PHTALOX^®^ solution (AM+PHTX Group) or 1 mL of PBS (C+ Group) was added and incubated in the absence of light for 43 min and 18 s. The initial 15 min were related to the incubation time in an incubator at 37 °C for the internalization of the PS, while the final 28 min and 18 s were related to the treatment at room temperature.

### 2.8. Antimicrobial Photodynamic Therapy

To perform aPDT, first, the PS was internalized, where the biofilms remained incubated with the PHTALOX^®^ solution for 15 min in the absence of light, followed by irradiation without removing the PS. For irradiation, a device with 54 LEDs (Biopdi / Irrad-Led 660–Brazil) with 70 mW of power, allowing for a uniform irradiance distribution and with light emission in the region of ± 660 nm, power density of 30 mW.cm^−2^ and energy density of 50 J.cm^−2^ was used as a light source. The irradiation time used was 28 min and 18 s, which was determined considering the irradiation parameters of the device [37].

#### 2.8.1. Obtaining the L Group

The same procedure as that for the C+ group described in item 2.6 was performed. However, after 15 min of incubation with PBS in the absence of light, aPDT was performed, following the parameters mentioned in item 2.8.

#### 2.8.2. Obtaining the AM+L Group

The same procedure described in item 2.6. was performed; however, after the AM fragment made contact with the bottom of the plate containing the biofilm, irradiation was performed, following the parameters mentioned in item 2.8.

#### 2.8.3. Obtaining the AM+aPDT Group

The same procedure described in item 2.7 was performed; however, in the final 28 min and 18 s of treatment, instead of remaining in the absence of light and at room temperature, aPDT was performed, following the parameters mentioned in item 2.8.

### 2.9. Analysis of Results

#### 2.9.1. Count of Colony-Forming Units

To perform the count of colony-forming units per milliliter (CFU/mL), the biofilm was detached by mechanical action, with the aid of disposable tips, and base 10 dilutions were made in PBS in the 24-well plate itself. A total of 10 μL of each well was plated on BHI Agar by the “drop method”, and the plates were incubated for 24 h at 37 °C. The colonies were quantified to obtain Log_10_ CFU/mL, and after that, the calculation of logarithmic reduction was performed, using the following formula:Logarithmic Reduction = Log_10_ (positive control) − Log_10_ (treated group)

#### 2.9.2. Metabolic Activity Test

A total of 2 mL of a 0.5 mg/mL solution of 3-(4,5-dimethyl-2-thiazolyl)-2, 5-diphenyl-2H-tetrazolium bromide (MTT, Invitrogen, Eugene, OR, USA) was added to the biofilms, followed by incubation in the absence of light at 37 °C for 3 h. Then, the reagent was removed from the wells and 1 mL of dimethylsulfoxide (DMSO, NEON, Suzano, SP, Brazil) was pipetted into the biofilms, followed by homogenization. Next, approximately 100 µL of DMSO, which had been in contact with the biofilm, was transferred to a 96-well plate for absorbance determination in a microplate reader (Biotek Elx808) under a wavelength of 570 nm [38]. The absorbance values found were converted into a percentage of metabolic activity in relation to the control. For this, the following formula was used:Metabolic Activity (%) = (Absorbance of groups − blank) × 100
(Control absorbance − blank)

Subsequently, the reduction percentage was calculated using the following formula:% Reduction = (% Metabolic Activity C + −% Metabolic Activity of treated group) × 100
% Metabolic Activity C+

#### 2.9.3. Statistical Analysis

Statistical analysis was performed using the GraphPad Prism 5 software. The different groups were analyzed using the Kolgomorov–Smirnov (KS) normality test, the One-Way ANOVA parametric test, and Tukey’s post-test, considering a significance level of 95% (*p* < 0.05).

### 2.10. Amniotic Membrane Analysis

#### Scanning Electron Microscopy

In order to visualize the integrity of the AM, the AMs from the groups AM, AM+L, AM+PHTX, and AM+aPDT were submitted to a scanning electron microscope (SEM). Thus, only treatments were carried out, without the contamination and assembly of the biofilm, in order to verify possible damage resulting only from the treatments tested. For this, the process described in 2.6 was carried out after the cryopreservation, and specific treatments for each group were performed. For fixation, a 2.5% glutaraldehyde + 2.5% paraformaldehyde + 0.05 mol cacodylate buffer was used for 2 h at 4 °C. After that, a wash was performed with 500 µL of PBS. Following the fixation, dehydration was performed with 5 min alcoholic baths in increasing concentrations of 30%, 50%, 70%, 100%, and 100%+HMDS. Drying was performed with the AM in 24-well plates at room temperature overnight. After that, the AMs were removed from the wells, metallized with a gold bath in a metallizer (Emitech, model K550X), and placed in the Scanning Electron Microscope (model EVO MA 10–ZEISS) for analysis.

## 3. Results

### 3.1. Evaluation by Counting Colony-Forming Units

With the methodology standardization tests, the influence of the pH of the final PHTALOX^®^ solution on the microbial growth during 43 min and 18 s of exposure was verified. There was no statistically significant difference (*p* < 0.05) regarding the number of CFU/mL of the C+ group, the biofilm in contact with PBS and the NaOH group, and the biofilm in contact with the 0.3% NaOH solution.

Table 2 indicates the mean values found for CFU/mL and Log_10_ reduction, based on the calculation of the logarithmic reduction of groups L, AM, AM+L, AM+PHTX, and AM+aPDT in relation to the control group of the *S. aureus* and *P. aeruginosa* strains.

The average CFU/mL of the strains (A) *S. aureus* and (B) *P. aeruginosa*, in relation to the groups C+, L, AM, AM+L, AM+PHTX, and AM+aPDT, are shown in Figure 2 on charts containing the Y-axis at log_10_.

Based on the statistical analysis of the results, it was possible to observe that, in both strains, there was no statistical difference between the C+ group and the L group or between the AM group and the AM+L group, demonstrating that the light used did not cause changes or damage to the AM. For the biofilms of both microorganisms, the AM, AM+L, AM+PHTX, and AM+aPDT groups showed significantly lower contagions (*p* < 0.05) when compared to the C+ group, demonstrating that the different treatments used were effective in the decrease in CFU/mL in the biofilms. When compared to each other, the AM+PHTX and AM+aPDT groups did not show a statistical difference for both strains, showing that the PS used was cytotoxic to bacterial cells, regardless of light.

### 3.2. Evaluation of Metabolic Activity

The absorbance values found were transformed into a percentage of metabolic activity, and the reduction percentages were calculated. These were demonstrated in the comparison of the groups L, AM, AM+L, AM+PHTX, and AM+aPDT in relation to the C+ group of the *S. aureus* and *P. aeruginosa* strains (Table 3).

The percentage means of the metabolic activity of the groups C+, L, AM, AM+L, AM+PHTX, and AM+aPDT are represented in Figure 3 in relation to the strains (A) *S. aureus* and (B) *P. aeruginosa*.

After the statistical analysis, it was observed that, despite a slight decrease in metabolic activity, there was no statistical difference between the C+ group and the L group or between the AM group and the AM+L group in both strains, demonstrating that the light used did not have relevant effects on the metabolic activity of the bacteria. In both strains, the AM, AM+L, AM+PHTX, and AM+aPDT groups showed lower metabolic activity (*p* < 0.05) when compared to the C+ group, demonstrating that the treatment with AM, associated with PHTX, and with aPDT caused a decrease in the metabolic activity of biofilms in all the treatments used. When compared to each other, the AM+aPDT group showed a significantly lower difference (*p* < 0.05) when compared to AM+PHTX in both strains, evidencing the decrease in metabolic activity with the association of AM, the PHTALOX^®^ solution, and photodynamic inactivation, indicating that this treatment is effective in reducing the metabolic capacity of these biofilms.

### 3.3. Analysis of the Amniotic Membrane by Scanning Electron Microscopy

The AM micrographs of the epithelial surface (Figure 4) and mesenchymal surface (Figure 5) were obtained by means of SEM (EVO MA 10–ZEISS), with the following parameters: Mag = 500 X, 1.00 K X, and 3.00 K X, respectively; EHT = 20.00 kV; WD = 9.5 mm; Signal A = SE1; I probe = 281 pA. The images obtained among the studied groups were compared: AM = Amniotic membrane without any interference other than fixation and metallization; AM+L = Amniotic Membrane exposed to light following the parameters of item 2.8.2.; AM+PHTX = Amniotic Membrane exposed to the PHTALOX^®^ solution, following the parameters of item 2.7.; AM+aPDT = Amniotic Membrane exposed to the PHTALOX^®^ solution and to light, following the parameters of item 2.8.3.

By analyzing the images obtained from the AM on its epithelial surface, it was possible to observe uniform polygonal epithelial cells in a mosaic pattern, with a well-defined intercellular junction and microvilli on the apical surface of the cell in the AM group. In the AM+L group, it was possible to observe that the epithelial cells remained intact and uniform when compared to the AM group; however, there was an increase in cytoplasmic bridges in the intercellular junction, and the microvilli were presented with less definition and as flattened.

By observing the AM+PHTX and AM+aPDT groups, a significant morphological alteration was observed. It is still possible to visualize the delimitation of epithelial cells; however, there was loss of intercellular junction space, resulting in a loss of definition. The microvilli were flattened in the AM+PHTX group, and the loss of definition of this structure was emphasized in the AM+aPDT group.

With the analysis of the images obtained from the AM on its mesenchymal surface, it was possible to observe the bundles of collagen fibers with homogeneous thickness and a dense appearance in the AM group. When viewing the images obtained from the AM+L group, it was possible to observe the bundles of collagen fibers with less homogeneous thickness and a less dense appearance. Observing the AM+PHTX and AM+aPDT groups, it is still possible to visualize the bundles of collagen fibers with less homogeneous thickness; however, in both groups, the loose and less compact aspect of the fibers was observed. In all analyzed groups, the surface presented a smooth appearance.

Macroscopically, a morphological difference was also reported between these groups, since the AMs with these treatments obtained a thinner and more fragile appearance, in addition to having acquired the greenish color of the PHTALOX^®^ solution, on both sides.

## 4. Discussion

The excellent results in controlling the proliferation of bacteria of great medical hospital relevance presented in the findings of this study corroborate the medical scientific literature regarding the evolution of microbial resistance, since, according to the WHO (2004), the issue of multi-resistant microorganisms with respect to antibiotics has the capacity to completely change the health system as we know it, unless this phenomenon is addressed and contained [39,40,41]. The scarcity of new antimicrobials that can replace those that have become ineffective demands the need to protect the effectiveness of existing agents and the creation of new efficient therapies [42,43].

The control of metabolic activity and the reduction in CFU/mL counts in *P. aeruginosa*, demonstrated in this work, are relevant and promising, since this bacterium is among the microorganisms that, in addition to having acquired resistance, are intrinsically resistant to more than one class of antimicrobial agents [42], whereas it often exhibits limited efficacy due to its adaptability and high resistance to common antimicrobials [44]. For a 3 Log_10_ reduction with PDT treatment, the study by Pérez-Laguna et al. (2020) used 16 µg.mL^−1^ of methylene blue as PS in *S. aureus* biofilms and 256 µg.mL^−1^ in biofilms of *P. aeruginosa*, demonstrating the resistance of this strain, since it was necessary to considerably increase the PS concentration in the *P. aeruginosa* treatments to obtain the same logarithmic reduction in both strains [45]. The present study, despite using higher concentrations of PS, used the same concentration for both bacteria tested, and the results obtained corroborate the intrinsic resistance of *P. aeruginosa*, because, even using a standard strain, an average growth was observed in the groups treated with AM+aPDT in CFU/mL of 6 Log_10_ greater than the results obtained with the bacteria *S. aureus*.

In this study, the same irradiation parameters were used for both microorganisms; however the study by Thakuri et al. (2011) suggested the need for a longer irradiation time in PDT to make *P. aeruginosa* cells unfeasible compared to *S. aureus*, associating this demand to the complex structure of the cell wall of Gram-negative species compared to Gram-positive ones [46]. These results agree with the results obtained, because although *P. aeruginosa* had a metabolic activity 4.08% lower than that obtained for *S. aureus*, after the AM+aPDT treatment, this decrease in metabolic activity was not enough to make the bacterial cells unfeasible, considering the recovery of CFU/mL obtained after the treatment of this strain.

Although the present study used bacterial cells in biofilm aggregates, this factor did not prevent the statistically significant decrease (*p* < 0.05) of the treated groups when compared to C+, reaching logarithmic reduction values of up to 10.41 with a decrease in metabolic activity of up to 69.30% for *S. aureus* and a logarithmic reduction of up to 5.82 with a decrease in metabolic activity of up to 73.38% for *P. aeruginosa*, in relation to the AM+aPDT treatment, which obtained the best response. Such results disagree with the study by Figueiredo-Godoi et al. (2022), in which the authors evaluated the action of PDT mediated by a new PS derived from chlorin e-6 (Fo-tenticine) on planktonic growth and biofilm in strains of *Acinetobacter baumannii*, obtaining a statistically significant decrease of about 2 Log_10_ in CFU/mL in the treatment of planktonic cells and no inhibition in the number of viable cells in the treatment of cells in the biofilm, demonstrating the resistance of this aggregate to antimicrobial photodynamic effects [47]. The positive effects on the control of the microbial biofilms of *S. aureus* and *P. aeruginosa* showed the gain to be even more important when considering that, in addition to the phenomena of acquired and intrinsic resistance, the capacity of biofilm formation is a relevant factor that confers resistance to the microorganisms. According to Saxena et al. (2019), bacterial biofilms show up to 1000 times greater resistance to antibiotics when compared to planktonic bacteria [48].

The results obtained in the present study on antimicrobial control in the use of AM agree with the study of Kjaergaar et al. (2001) that analyzed the antibacterial potential of fresh AM in clinical isolates of *Streptococcus* group A of Lance-field, *S. aureus*, *S. saprophyticus*, *Enterococcus faecalis*, *E. coli*, *P. aeruginosa*, *Acinetobacter calcoaceticus*, and *Lactobacillus* sp. [3]. Similar to the findings of this work, the authors kept the AM in contact with the microorganisms for 24 h, although they used planktonic cells and not biofilm. The exposure time was identical, however, to the analysis of the study of Kjaergaar et al. (2001), which was qualitative due to the production of halo and not quantitative, like the findings of the present work. It is noteworthy that, despite being a qualitative work, after 24 h of incubation, the researchers removed the AM and incubated the plates for another 24 h, and no reversion of the restriction was observed, suggesting a bactericidal effect of the AM.

The use of membranes in cryopreservation at –80 °C for 10 days in the carried out research does not corroborate the results reported by Kjaergaar et al. (2001), who used fresh AM [3]. Furthermore, the findings demonstrated here revealed the action of all AMs enrolled from different sources, in line with Tehrani, Ahmadiani, and Niknejad (2013), who compared the antimicrobial efficiency of fresh AM, cryopreserved AM, and lyophilized AM against the strains of *S. aureus, P. aeruginosa*, and *E. coli* and observed a better antimicrobial effect (88%) on fresh membranes and a lesser effect on cryopreserved (59%) and lyophilized (62%) AMs, also using the stapes halo formation as the antimicrobial evaluation [49]. These results can be clarified considering the results obtained in the study by Pogozhykh et al. (2018), who observed, from the MTT assay, a decrease in the metabolic activity of AM cells after cryopreservation, as compared to fresh AM [50]. In addition, Tehrani, Ahmadiani, and Niknejad (2013) also measured the amount of elafin in the AM extracts in these three processing methods, resulting in a decrease in the amount of elafin in the AMs that underwent the cryopreservation and lyophilization methods [49]. The antimicrobial effect, although the lowest among the tested processing groups, was still present in the cryopreserved AMs, suggesting that the antibacterial properties of AM are not only related to elafin but also to other components of the extracellular matrix [49], which explains the fact that the present study used the same storage method and, based on the treatment with isolated AM, still achieved values of logarithmic reduction of 1.76 and 0.76 and a decrease in the metabolic activity of 40.19% and 25.88% of *S. aureus* and *P. aeruginosa* biofilms, respectively. It is noteworthy that the present study standardized the orientation of the AM, keeping the mesenchymal surface in contact with the biofilms, but another study by Tehrani et al. (2017) demonstrated that the orientation of the AM (epithelial or mesenchymal surface in contact with the microorganism) does not affect the antimicrobial efficiency in the direct contact method [51].

By using SEM analysis, the study by Modaresifar et al. (2017) found that the AM stored by the cryopreservation method is kept intact, without any evidence of rupture or tear, and with a uniform layer of epithelial cells [52], agreeing with the findings of the MA group in the present study. The authors also indicated the preservation of its five layers, including the epithelial layer, basal membrane, compact layer, fibroblast layer, and spongy layer, as well as in fresh AM, even after cryopreservation. The morphological alterations presented by the AM with the SEM analysis after the treatments of the AM+PHTX and AM+aPDT groups corroborate the study by Ab Hamid et al. (2014), who evaluated the AM after the effects of gamma radiation in different preservation methods [53]. The authors analyzed fresh and cryopreserved AM, and both showed morphological alterations after gamma irradiation, with findings similar to those of the present study, such as a loss of the definition of epithelial cells, flattened microvilli, a reduction in the intercellular channel, and thinner bundles of collagen fibers. The use of different techniques has been reported to decontaminate AM for clinical use, and the study by Wehmeyer et al. (2015) used a new technique involving supercritical carbon dioxide and also evaluated post-treatment AM integrity by SEM, resulting in a flattened epithelium with less distinguishable cell boundaries compared to untreated AM [54], strengthening the findings of the present study regarding the flattening of epithelial cells/microvilli and the loss of intercellular junction space in the AM+PHTX and AM+aPDT groups. Considering the logarithmic reduction values achieved and the decrease in the metabolic activity of *S. aureus* and *P. aeruginosa* biofilms, it can be considered that the morphological alteration observed through SEM did not hinder the antimicrobial action of AM nor its action synergy with the other tested treatments.

Despite the relevant results obtained from the treatment of biofilms with AM, the experimental group that achieved the greatest logarithmic reduction and decrease in metabolic activity in the present study was the association of AM with aPDT (AM+aPDT). As a hypothesis to be further investigated, considering the excellent antimicrobial potential resulting from the synergism of these two treatments, there is a possibility that AM plays a surfactant role, causing a slight breakdown of the PHTALOX^®^ solution and improving its photodynamic action, considering that the phthalocyanines exhibit a high aggregation tendency, and the PS that form aggregates have a lower photosensitization efficiency, as aggregation reduces the excitation lifetime of the molecule [55,56,57]. The surfactant capacity of AM has already been demonstrated in the study by Lemke et al. (2017), in which the authors identified that the cells that make up the AM contain lamellar bodies and express all four surfactant proteins, in addition to having essential lipid species found in the pulmonary surfactant in both amniotic sub regions [58]. Furthermore, it showed the ability to reduce surface tension, like the human lung surfactant, strengthening the hypothesis that AM played such a role in the present study. However, it is not possible to state that 15 min of incubation would be sufficient for this action, suggesting further studies for better investigation. In addition to enhancing the results arising from the surfactant capacity of MA, another possible way to intensify the results would be to include PHTALOX in bioactive nanomaterials, serving as PS transporters to increase the therapeutic effects of aPDT, as demonstrated in the study by QI et al. (2019). The study supported that these new applications of aPDT not only have therapeutic potential of oral plaque-initiated diseases but also have broad applicability to other biomedical and tissue engineering applications [59].

The promising antimicrobial effects demonstrated by AM add to the other biological properties already demonstrated such as the production of anti-inflammatory agents such as hyaluronic acid, the suppression of pro-inflammatory cytokines, anti-fibrotic properties due to the negative regulation of transforming growth factor (TGF-β), low antigenicity, and immunomodulatory properties as a result of the factor secreted by epithelial cells, which, during pregnancy inhibit the migration of macrophages and natural killer cells in order to prevent a maternal immune attack [2,60,61,62]. Considering the lack of statistical significance between the AM+PHTX and AM+aPDT groups regarding the decrease in the number of CFU/mL, it is possible to consider that the best therapy was the one adopted in the AM+PHTX experimental group, because, by associating AM and the PHTALOX^®^ solution, it was possible to obtain logarithmic reduction values of 8.42 and 4.79 and a decrease in metabolic activity of 56.02% and 54.08% of the biofilms *S. aureus* and *P. aeruginosa*, respectively, without requiring irradiation to obtain significant results. Whereas the isolated treatment with the PHTALOX^®^ solution and that with AM are already used in clinical studies for the treatment of wounds and ulcers [63,64,65,66,67], their association may emerge as a strategic alternative for the treatment of these infected wounds, since this synergistic effect promoted a significant decrease in the number of microorganisms, even in their biofilm state, and this is due to the high regenerative potential of AM, which reduces the use of antibiotics and interferes with the non-chronicity of an infected wound, encouraging the continuation of this study for the in vivo method.

## 5. Conclusions

The treatment using AM showed promising results as an alternative therapy for controlling S. aureus and P. aeruginosa biofilms, with an even more significant anti-biofilm effect associated with the PHTALOX^®^ solution. Furthermore, although the SEM analysis on both sides of the AM showed morphological alterations emphasized in the AM+PHTX and AM+aPDT groups, such alterations did not hinder its antimicrobial effect. These findings show the possibility of using this treatment to control biofilms, especially where their presence can cause significant problems, such as wounds.

## Figures and Tables

**Figure 1 jfb-14-00151-f001:**
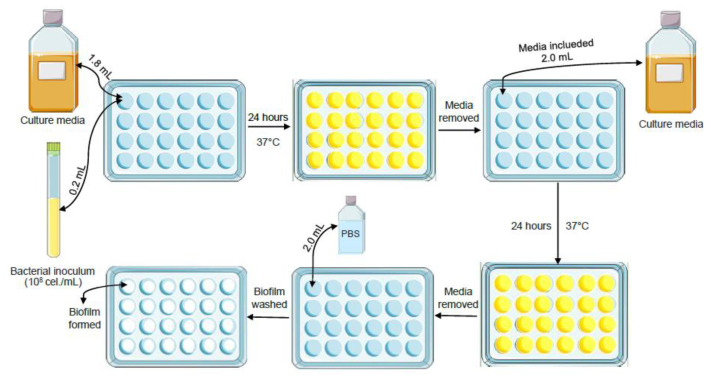
Biofilm preparation and assembly scheme.

**Figure 2 jfb-14-00151-f002:**
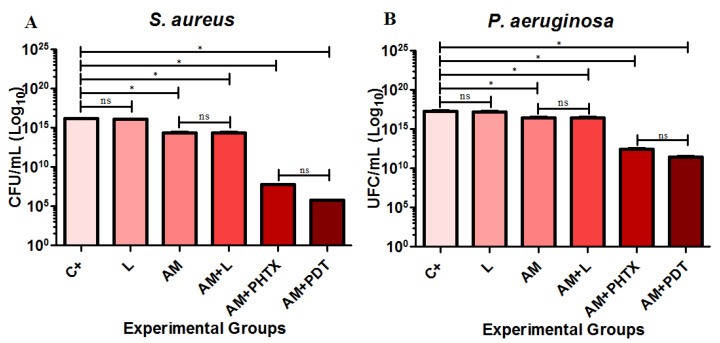
Determination of the average values of CFU/mL of the biofilms of the strains: (**A**) *S. aureus* and (**B**) *P. aeruginosa*, C+, L, AM, AM+L, AM+PHTX, and AM+aPDT, respectively. The symbol * identifies the statistical difference of *p* < 0.05 between the groups, and the acronym ns identifies the non-statistical significance.

**Figure 3 jfb-14-00151-f003:**
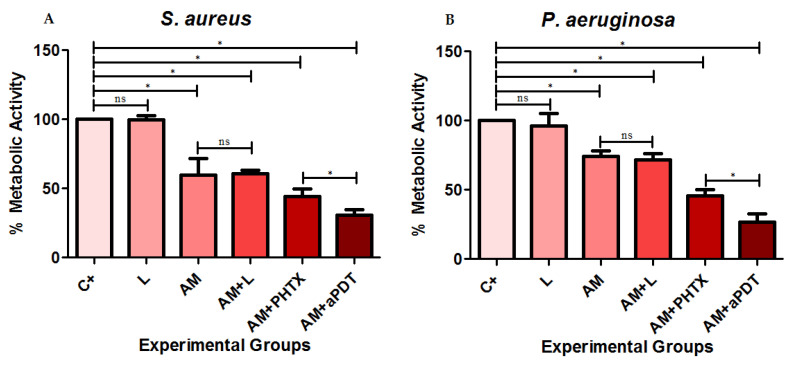
Determination of the average percentage of the metabolic activity of the biofilms of the strains (**A**) *S. aureus* and (**B**) *P. aeruginosa* in relation to the groups C +, L, AM, AM+L, AM+PHTX, and AM+aPDT, respectively. The symbol * identifies the statistical difference of *p* < 0.05 between the groups, and the acronym ns identifies the non-statistical significance.

**Figure 4 jfb-14-00151-f004:**
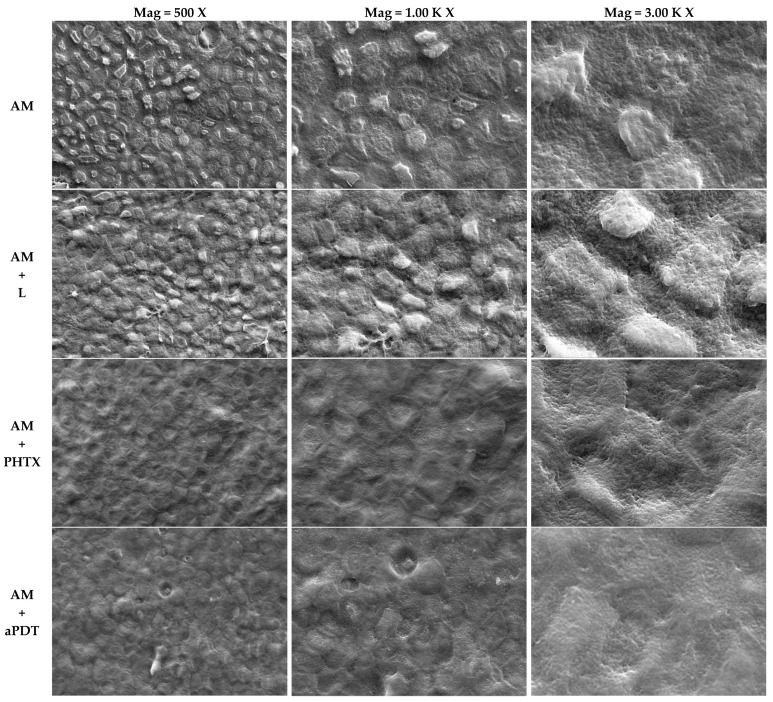
Micrographs obtained by scanning electron microscopy of the epithelial surface of the AM after the treatments of the groups AM, AM+L, AM+PHTX, and AM+aPDT. Comparison of images obtained between the studied groups Group AM (AM treatment), Group AM+L (AM and light treatment), Group AM+PHTX (AM and PHTALOX^®^ treatment), and Group AM+aPDT (AM and PHTALOX^®^ treatment, with exposure to light) by scanning electron microscopy (EVO MA 10–ZEISS) with the following parameters: Mag = 500 X, 1.00 K X, and 3.00 K X, respectively; EHT = 20.00 kV; WD = 9.5 mm; Signal A = SE1; I probe = 281 pA.

**Figure 5 jfb-14-00151-f005:**
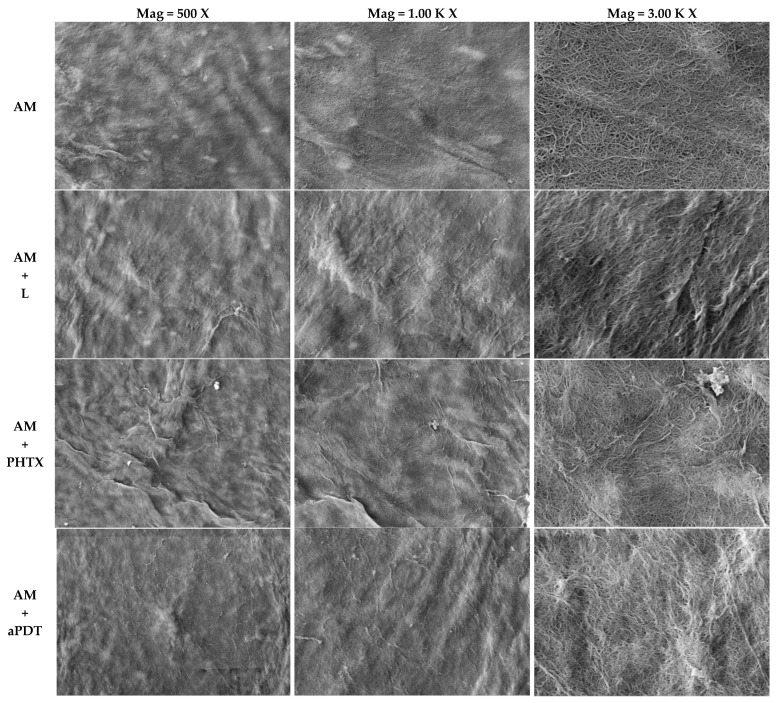
Micrographs obtained by scanning electron microscopy of the mesenchymal surface of the AM after the treatments of the groups AM, AM+L, AM+PHTX, and AM+aPDT. Comparison of images obtained between the studied groups Group AM (AM treatment), Group AM+L (AM and light treatment), Group AM+PHTX (AM and PHTALOX^®^ treatment), and Group AM+aPDT (AM and PHTALOX^®^ treatment, with exposure to light) by scanning electron microscopy (EVO MA 10–ZEISS) with the following parameters: Mag = 500 X, 1.00 K X, and 3.00 K X, respectively; EHT = 20.00 kV; WD = 9.5 mm; Signal A = SE1; I probe = 281 pA.

**Table 1 jfb-14-00151-t001:** Scheme of experimental groups. Group C+ (no treatment), Group L (light treatment), Group AM (AM treatment), Group AM+L (AM and light treatment), Group AM+PHTX (AM and PHTALOX^®^ treatment), Group AM+aPDT (AM and PHTALOX^®^ treatment, with exposure to light).

	AM	PHTALOX^®^	IRRADIATION
Group C+			
Group L			X
Group AM	X		
Group AM+L	X		X
Group AM+PHTX	X	X	
Group AM+aPDT	X	X	X

**Table 2 jfb-14-00151-t002:** Determination of mean values of CFU/mL and logarithmic reduction in biofilms of the strains *S. aureus* and *P. aeruginosa*, in relation to the groups C+, L, AM, AM+L, AM+PHTX, and AM+aPDT.

	*S. aureus*		*P. aeruginosa*	
	CFU/mL	Log_10_ Reduction	CFU/mL	Log_10_ Reduction
Group C+	1.5 × 10^16^	-	1.8 × 10^17^	-
Group L	1.3 × 10^16^	0.06	1.7 × 10^17^	0.02
Group AM	2.6 × 10^14^	1.76	3.1 × 10^16^	0.76
Group AM+L	2.5 × 10^14^	1.78	3.1 × 10^16^	0.76
Group AM+PHTX	5.7 × 10^7^	8.42	2.9 × 10^12^	4.79
Group AM+aPDT	5.9 × 10^5^	10.41	2.7 × 10^11^	5.82

**Table 3 jfb-14-00151-t003:** Determination of the mean values of the percentage of metabolic activity and the percentage of the reduction in biofilms of the strains *S. aureus* and *P. aeruginosa*, in relation to the groups C +, L, AM, AM+L, AM+PHTX, and AM+aPDT.

	*S. aureus*		*P. aeruginosa*	
	% Metabolic Activity	% Reduction	% Metabolic Activity	% Reduction
Group C+	100	-	100	-
Group L	99.68	0.32	95.78	4.22
Group AM	59.81	40.19	74.12	25.88
Group AM+L	60.88	39.12	71.70	28.30
Group AM+PHTX	43.98	56.02	45.92	54.08
Group AM+aPDT	30.70	69.30	26.62	73.38

## Data Availability

All data obtained or analyzed during the present study are included in the article.

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
