# Peer review of "Antimicrobial Effect of the Amniotic Membrane Isolated and Associated with Photodynamic Therapy"

_jfb, 2023, doi:10.3390/jfb14030151_

Round 1
Reviewer 1 Report
The authors aimed to evaluate the antimicrobial effect of AM isolated and associated with aPDT using PHTALOX as a photosensitizer against biofilms. The manuscript has been well-written and falls within the scope of the journal. However, some major concerns should be addressed before this manuscript is accepted for publication.
Comments:
1. What is the relevance between morphologic change and antimicrobial efficacy?
2. Ln 448: “In addition to the antimicrobial effect, the use of AM promotes the production of anti-inflammatory agents such as hyaluronic acid, the suppression of pro-inflammatory cytokines, anti-fibrotic properties due to negative regulation of transforming growth factor (TGF-β), low antigenicity and immunomodulatory properties as a result of the factor secreted by epithelial cells, which during pregnancy inhibits the migration of macrophages and natural killer cells in order to prevent a maternal immune attack.”
These findings were already available, and the authors should not claim their materials are superior based on the previously published data. The authors should still need experimental data to support this point of view compared to those aPDT group. Additionally, the authors should provide more in-depth discussion, including novel findings and/or approaches compared to those already published works.
3. Section 4, Ln 317: Regarding recent findings and advances combating microorganisms, authors need to refer to more comprehensive and recent literature:
10.2147/IJN.S212807
Author Response
Response to REVIEWER 1
- What is the relevance between morphologic change and antimicrobial efficacy?
Response: Scanning electron microscopy (SEM) was performed in order to certify the integrity of the AM, in this way, if profound changes were verified in the epithelial layer, such as degeneration and cell loss, linked to a lower antimicrobial effect of the AM associated with other treatments (PHTX and PDTa), it could be confirmed that the morphological alteration would be relevant for the antimicrobial effect. However, this did not occur in the present study, since the AM, despite having presented morphological differences, these did not prevent its antimicrobial potential, taking into account the results obtained. This SEM analysis was based on studies showing that different AM protocols and storage may affect the antimicrobial effect (https://doi.org/10.3389/fmicb.2020.00469).
- Ln 448: “In addition to the antimicrobial effect, the use of AM promotes the production of anti-inflammatory agents such as hyaluronic acid, the suppression of pro-inflammatory cytokines, anti-fibrotic properties due to negative regulation of transforming growth factor (TGF-β), low antigenicity and immunomodulatory properties as a result of the factor secreted by epithelial cells, which during pregnancy inhibits the migration of macrophages and natural killer cells in order to prevent a maternal immune attack.” These findings were already available, and the authors should not claim their materials are superior based on the previously published data. The authors should still need experimental data to support this point of view compared to those aPDT group. Additionally, the authors should provide more in-depth discussion, including novel findings and/or approaches compared to those already published works.
Response: We agree with the suggestion and changes were made in text for greater clarity.
- Section 4, Ln 317: Regarding recent findings and advances combating microorganisms, authors need to refer to more comprehensive and recent literature: 10.2147/IJN.S212807
Response: We appreciate the suggestion and the literature has been considered and added to the discussion in the paper.

Reviewer 2 Report
Review Report
I would like to thank all authors of the manuscript for their good and novelty manuscript titled as (Antimicrobial effect of the amniotic membrane isolated and associated with photodynamic therapy) which submitted to journal (Journal of Functional Biomaterial) JFB (ISSN 2079-4983)) .
1-The manuscript is original and novel as it aims to the treatment using AM as an alternative therapy to control S. aureus and P. aeruginosa biofilms
2-The Presentation of the manuscript is good which attract the Interest to the readers.
3-Minor revision is needed to English language and style.
4-The introduction provide sufficient background and include all relevant references and styled according to the style of the journal.
5- All the cited references relevant to the research.
6- All the cited references relevant to the research.
7- The methods adequately described.
8- The results clearly presented.
9- The conclusions supported by the results.
Corrections are highlighted in the pdf file
So, I recommend accepting after minor revision (corrections to minor methodological errors and text editing.

Author Response
Response to REVIEWER 2
1- The manuscript is original and novel as it aims to the treatment using AM as an alternative therapy to control S. aureus and P. aeruginosa biofilms
2-The Presentation of the manuscript is good which attract the Interest to the readers.
3-Minor revision is needed to English language and style.
4-The introduction provide sufficient background and include all relevant references and styled according to the style of the journal.
5- All the cited references relevant to the research.
6- All the cited references relevant to the research.
7- The methods adequately described.
8- The results clearly presented.
9- The conclusions supported by the results.
10- Corrections in the pdf file
Response: We appreciate all the comments. We agree with the suggestions and corrections presented in the PFD file, and changes were made in text for greater clarity. In topic 2.6 and 2.7 there is no reference since the protocol was established and standardized in our laboratory, so no reference has been added to these topics.

Reviewer 3 Report
The abstract is unclear and needs to revise. Include more methodology parts clearly and concisely.
In the introduction, no research data is shown as to what will be the benefit or what will be a contribution of this study to the existing study and knowledge.
Add the description and explanation of all abbreviations for Table 1. Scheme of experimental groups.
Over all the work is new and can be helpful in microbial therapy.
1) In figure 3 and 4, the scale is missing 2) The main question addressed is to evaluate the antiimicrobial effect of AM isolated and associated with aPDT using the PHTALOX® as a photosensitizer (PS) against Staphylococcus aureus and Pseudomonas aeruginosa biofilms. The effect of AM is evaluated for antibacterial effect. 3) Do you consider the topic original or relevant in the field? Does it address a specific gap in the field - The study is not novel as many studies have shown the antibacterial effect of phthalocyanine and PDT effect. 4) What specific improvements should the authors consider regarding the methodology? What further controls should be considered? -Give the biofilm preparation schematic diagram, the characterization of biofilm is missing and application in general. The toxicity of biofilm should be done. The bacterial assay test using an agar plate should be performed. 5) Conclusion is not properly written, future prospects and results are not discussed in a way that reflect that study is showing positive response.
Author Response
Response to REVIEWER 3
The abstract is unclear and needs to revise. Include more methodology parts clearly and concisely.
Response: We appreciate the suggestion, however the journal limits the abstract to 200 words, and the current abstract has already reached that number. Therefore, as far as possible, changes were made to improve the clarity of the text.
In the introduction, no research data is shown as to what will be the benefit or what will be a contribution of this study to the existing study and knowledge.
Response: We appreciate the suggestion and changes were made in the introduction.
Add the description and explanation of all abbreviations for Table 1. Scheme of experimental groups.
Response: We agree with the suggestions and changes were made for greater clarity and description in text.
Over all the work is new and can be helpful in microbial therapy.
Response: We appreciate.
1) In figure 3 and 4, the scale is missing
Response: We appreciate the suggestion, but the scale used for the analysis with the scanning electron microscope shown in figures 3 and 4 are mentioned in the first paragraph of the item 3.3, however the parameters were also added in the description of each image for greater clarity.
2) The main question addressed is to evaluate the antiimicrobial effect of AM isolated and associated with aPDT using the PHTALOX® as a photosensitizer (PS) against Staphylococcus aureus and Pseudomonas aeruginosa biofilms. The effect of AM is evaluated for antibacterial effect.
Response: We appreciate the comment.
3) Do you consider the topic original or relevant in the field? Does it address a specific gap in the field - The study is not novel as many studies have shown the antibacterial effect of phthalocyanine and PDT effect.
Response: I consider the topic original and relevant, because although there are already many works proving the antimicrobial effect of photodynamic therapy and phthalocyanine, the study is a pioneer in the association of this therapy with the amniotic membrane, even more so using a new commercial phthalocyanine as a photosensitizer against bacteria in their biofilm state.
4) What specific improvements should the authors consider regarding the methodology? What further controls should be considered? -Give the biofilm preparation schematic diagram, the characterization of biofilm is missing and application in general. The toxicity of biofilm should be done. The bacterial assay test using an agar plate should be performed.
Response: We appreciate the suggestions, however in all the tests a control group (C+) was inserted, which served as the basis for all the calculations shown in the results topic. The microorganisms used, the preparation of the inoculum and the assembly of the biofilm are described in detail in items 2.4, 2.4.1 and 2.4.2, respectively. The schematic diagram of biofilm preparation was prepared and inserted in the text (Figure 1). Biofilm toxicity was performed using the MTT test and is described in item 2.9.2. The test using the agar plate count was also performed and is described in item 2.9.1.
5) Conclusion is not properly written, future prospects and results are not discussed in a way that reflect that study is showing positive response.
Response: We agree with the suggestions and changes were made in the conclusion for greater clarity.

Round 2
Reviewer 1 Report
Dear authors,
Thank you for the comprehensive revisions. Congratulations on your excellent work!